# A Multi-Objective Optimization Method for the Design of a Sustainable House in Ecuador by Assessing LCC and LCEI

Yuan Chen and Stephanie Gallardo *

School of Civil Engineering, Zhengzhou University, Zhengzhou 450001, China; chen_yuan@zzu.edu.cn
* Correspondence: ssgallardo@gs.zzu.edu.cn

**Abstract:** The building industry significantly contributes to global warming, driving the demand for sustainable construction and green buildings. However, barriers like cost concerns and limited knowledge persist. Previous studies have used multi-objective optimization (MOO) to minimize life cycle cost and environmental impact, often emphasizing energy efficiency. In equatorial climates, unique factors like material selection must be considered. This study assesses the cost-effectiveness of sustainable materials, focusing on envelope materials in Ecuador. The case study is a single-family house in the equatorial climate, optimized using Building Information Modeling (BIM), Life Cycle Assessment (LCA), and Life Cycle Cost Analysis (LCCA). In this study, a MOO process using the weighted sum approach (WSA) identifies sustainable house designs. The sustainable houses achieve a 98% decrease in Ozone Depletion Potential, a 75% reduction in Global Warming Potential, and a 45% drop in Primary Energy Demand, although they still incur a 30% increased cost. The results offer a foundation for cost-effective, eco-friendly housing solutions. Bamboo emerges as a promising material with local acceptance. This research highlights the significance of material selection in sustainable construction and provides a replicable approach for diverse settings. It aims to promote sustainable housing solutions in Ecuador and beyond.

**Keywords:** sustainable construction; green building materials; multi-objective optimization; environmental impacts; life cycle cost

## 1. Introduction

The construction industry plays a significant role in environmental degradation and global warming [1]. The focus is shifting toward green construction and sustainable materials to combat these issues. Nevertheless, cost concerns and limited knowledge among decision-makers are key barriers to adopting sustainable construction [2–6]. Therefore, the rising demand for sustainability highlights the need to address economic challenges [7]. Achieving innovative and affordable sustainable designs involves considering the entire building's life cycle. It is crucial to reduce the LCEI (Life Cycle Environmental Impact) of conventional constructions and the LCC (Life Cycle Cost) of green buildings, with a focus not just on the construction phase but also on the operational phase, which often accounts for over 80–85% of the total life cycle energy [8].

Evaluating a building's environmental footprint and cost implications throughout its lifespan requires a comprehensive assessment of all components and life cycle phases [9]. Approaches such as LCA (Life Cycle Assessment) and LCCA (Life Cycle Cost Analysis) are robust tools used to evaluate both environmental and economic performance, covering their complete life cycle, from material extraction to demolition, and accounting for operational and embodied aspects. Integrating Building Information Modeling (BIM), LCA, and LCC offers a holistic perspective. It shifts the decision-making process away from merely considering initial investment costs when selecting materials or systems. Instead, it encompasses operational expenses and, notably, the associated environmental consequences.

A significant body of research has approached the optimization of the cost and environmental impacts of buildings. Certain advanced studies have employed mathematical

models to achieve multi-objective optimization (MOO), with a specific emphasis on reducing energy consumption during a building's operational phase. However, the impact of energy consumption is more pronounced in buildings reliant on HVAC (Heating, Ventilation, and Air Conditioning) systems, especially in regions with extreme weather conditions during the winter and summer seasons.

This situation contrasts with countries located over the equatorial line, where the climate remains stable year-round, resulting in energy consumption primarily from lighting and appliances. In Ecuador, the priority moves to material selection in minimizing costs and environmental impacts because of the stable equatorial climate.

This study highlighted three substantial research gaps: (1) The need for an integrated approach that combines BIM-LCA tools and LCCA. This integration can assist decision-makers in identifying design strategies, materials, and technologies that strike a balance between environmental impact and construction costs. (2) The importance of adapting house designs to the unique climate conditions of Ecuador, a developing country characterized by its biodiversity in the equatorial climate. Such designs should consider not only energy efficiency but also local materials and construction techniques appropriate for this specific context. (3) The absence of a concise optimization methodology that bridges the gap between research and practical implementation, particularly in residential projects. Developing such a methodology is essential for the actual adoption of sustainable and cost-effective building practices.

This study utilized a MOO approach to minimize both LCC and LCEI metrics, using a conventional low-cost house as the baseline. Sustainable materials and practices specific to the Ecuadorian construction industry context were identified in the optimization process. Additionally, the use of energy-efficient photovoltaic panels was considered, taking advantage of the consistent sunlight availability in the equatorial climate.

This research endeavors to report the identified research gaps by advancing the optimization of single-family house designs in Ecuador. The integration of BIM, LCA, and LCC facilitated the creation of an optimized house that considers both cost-effectiveness and minimal environmental impact. Building upon prior studies using MOO in conjunction with the WSA, this study benefits from Ecuador's distinct energy profile and aims to translate knowledge into actionable solutions.

## 2. Background

### 2.1. Environmental Analysis

An environmental impact assessment is crucial for evaluating the environmental repercussions associated with a building's construction and operation. It involves identifying, assessing, and mitigating potential adverse environmental impacts, aiming for a comprehensive understanding and management of environmental considerations throughout the entire life cycle of a project.

Environmental analysis approaches in construction involve various methodologies aimed at understanding and mitigating the environmental impact of construction processes. The adoption of BIM warrants amplification given its multifaceted integration with various project aspects, notably its pivotal role in environmental impact assessment. Green Building Information Modeling (Green BIM) integrates sustainable design and construction practices into a digital modeling framework, facilitating the identification and assessment of environmentally friendly options during the project life cycle. A study conducted in Peru [10] revealed a noteworthy surge in the BIM market from 2017 to 2020, particularly in the domain of extracting 2D drawings and quantity take-off. Despite these advancements, there remains a lag in BIM adoption among designers. Policymakers should cultivate an enabling environment for collaborative digital project delivery, positioning it as a strategic tool to propel the green building industry forward.

Life Cycle Assessment (LCA) assesses the environmental impacts of a construction project or product from cradle to grave, considering resource extraction, production, use, and disposal. The present study combines BIM and LCA, recognizing its scope in evaluating

the environmental impact of projects throughout their entire life cycle. LCA extends beyond material flows, encompassing considerations such as energy consumption details, emission data, and material quantities. This diverse set of data can be efficiently sourced from 3D models or energy simulations linked to BIM, exemplifying the synergistic potential of integrating these two methodologies in construction projects.

Indeed, other approaches also play integral roles in advancing green construction practices. Material Flow Analysis (MFA) examines the flow and transformation of materials within construction processes, helping to identify opportunities for resource efficiency and waste reduction. A study conducted in China employing MFA sheds light on material demand and environmental impact emanating from buildings [11]. The analysis discerned material flows pertaining to steel, cement, brick, and other crucial components. While MFA provides a comprehensive understanding of material consumption, it should be noted that it does not directly yield data regarding $CO_2$ emissions or embodied energy. Instead, MFA proves valuable when the individual impact of each material is discerned. For instance, it aids in recognizing the benefits delivered from recycling practices. This approach thus serves when the specific impacts of each material are well established.

Lean construction, a project management philosophy, focuses on optimizing construction processes, eliminating waste, and enhancing overall efficiency to reduce environmental impacts and improve project delivery. Lean construction contributes to green construction objectives by curbing material waste and enhancing economic benefits. In Latin America, particularly highlighted in a study from Colombia, the full adoption of lean construction practices has yet to gain substantial traction [12]. Notably, Brazil and Chile stand out for their strides in developing advanced technologies aimed at boosting productivity in the construction sector. This strategy should be replicated by other countries from the region as it aligns with the core principles of sustainability.

These diverse approaches offer a comprehensive toolkit for promoting sustainability in construction, integrating considerations from design to implementation and beyond.

Green BIM

Green building materials are materials that align with ecological principles, promote health and well-being, incorporate recycled content, and exhibit high-performance attributes. They are central to material selection, addressing sustainability pillars that encompass environmental, economic, and social benefits [13]. Green materials use the Earth's resources responsibly, including renewable plant materials like bamboo and straw; dimension stone; recycled stone; recycled metal; and other products known for their non-toxic, renewable, and recyclable properties. To further enhance sustainability, it is advisable for building materials to be sourced and manufactured locally, reducing the energy expended on transportation [14].

BIM is invaluable for designers in assessing diverse design alternatives during the conceptual phase of a building's life cycle, emphasizing energy considerations. Increasingly, BIM is being embraced as a tool for predicting and monitoring the environmental implications of construction. Designers can use BIM to select building materials thoughtfully, which has an influence on a building's operational energy consumption and costs. This allows owners and designers to make energy-related decisions with a profound impact on the projected building's life cycle cost [15].

The concept of Green BIM integrates green building principles with BIM technology to enhance sustainability and energy efficiency in construction projects. Green BIM guides stakeholders in adopting sustainable techniques while maintaining sustainability goals. Many projects that utilize Green BIM technology have achieved favorable outcomes by effectively balancing sustainability and economic considerations [15]. BIM can be a practical tool for sustainability analysis, encompassing factors like building orientation, envelope design, and construction materials.

## 2.2. Ecuadorian Context

Ecuador's unique geographic location at the equator in South America results in diverse climatic zones, including the Amazon rainforest, the Andes Mountains, and the coastal region. The methodology proposed in this study applies to the entire country, excluding the Paramo and temperate climates in the highlands. The coastal and Amazon rainforest regions, characterized by a tropical climate with consistent and mild temperatures ranging between 23 °C and 26 °C, offer wide options for construction materials and design considerations. However, the vast biodiversity in Ecuador is particularly vulnerable to the detrimental effects of climate change. The predominant manifestations include heightened occurrences of droughts and infrequent yet intensified rainfall events [16].

Concrete is the dominant building material in Ecuador, with 93% of all the constructions featuring concrete structures according to the 2015 national census. Within these, residential buildings represent 88%, underlining the importance of concrete in the housing sector [13]. Roofing materials also align with this trend, where reinforced concrete makes up approximately 59% of roofing structures, while fiber-cement panels and zinc-alloy roofs hold substantial shares at 23% and 11%, respectively. This diverse usage highlights factors like cost-effectiveness, weather resistance, and longevity in influencing roofing material selection. For wall assemblies, concrete blocks are used in about 62% of wall constructions, and clay bricks constitute a significant 36% share, demonstrating the versatility and adaptability of concrete in Ecuador's construction industry.

Ecuador's location over the equator results in consistent weather conditions throughout the year, reducing the need for space heating or air conditioning in most regions. The residential sector accounts for a significant portion of national electricity demand, comprising 36% of the total demand, and offers promising opportunities for energy conservation. Efforts have been made over the past decade to integrate renewable energy sources and enhance energy efficiency [17] with solar thermal systems or photovoltaic (PV) panels on rooftops, benefiting from Ecuador's stable sunlight conditions.

A critical concern in residential construction in Ecuador revolves around social housing initiatives. The governmental project "Houses for Everyone", initiated by the Ministry of Urban Development and Housing in 2019, aims to promote affordable dwellings for the vulnerable population and improve living conditions [18]. This study focuses on "Houses for Everyone" because it tailors house designs to Ecuadorian conditions nationwide, considering climate variations, social requirements, construction methods, materials availability, structural safety, and more.

## 2.3. Integration of BIM, LCA, and LCCA

Integrated BIM represents opportunities to advance sustainability within the Architecture, Engineering, and Construction (AEC) industry. One notable benefit is its efficiency in data collection, making LCA and LCCA more streamlined. These advantages provide decision-makers with a more accurate and informed approach to assessing the environmental impacts of construction projects [19].

LCA is a systematic methodology used to evaluate the environmental performance of material assemblies and entire structures throughout their entire lifespans. It considers all stages, from raw extraction and production to transportation, construction use, maintenance, and eventual disposal or recycling. LCA offers a comprehensive understanding of the sustainability and environmental implications of a construction project, allowing for a holistic analysis from an ecological standpoint. BIM-LCA tools provide digital representations of buildings incorporating LCA methodologies [20–23]. The process of conducting LCA using BIM involves creating a BIM model, defining the scope, identifying and quantifying environmental impacts, conducting sensitivity analysis, and evaluating results.

LCCA, on the other hand, assesses the total cost of a project over its entire life cycle, considering various cost components from the initial investment to disposal. It recognizes the importance of optimizing the entire LCC for making informed decisions during the

design and planning stages [24,25]. Focusing solely on upfront expenses has been a common practice during the design phase of housing projects [26]. LCCA promotes cost-effective choices that minimize long-term operational and maintenance expenses [27,28], making it valuable for assessing green building initiatives, cost reduction strategies, and alternative comparisons.

ISO 15865 [29] establishes clear terminology and methodology for LCCA to promote its widespread adoption within the construction industry. It provides a structured framework for consistent LCC predictions and performance evaluation, facilitating more rigorous comparative analyses and cost benchmarking. Within LCCA, various approaches exist to assess the economic aspects of a project [30], with Net Present Value (NPV) being the most employed tool [31]. NPV assesses the economic feasibility of projects, accounting for the time value of money and determining the present value of all expected cash flows, including costs and benefits, throughout the building's life cycle.

The integration of LCA and LCC represents a comprehensive approach to evaluating the environmental and economic aspects of a project throughout its whole life cycle. This harmonious combination empowers decision-makers to make well-informed choices that consider both environmental sustainability and cost-effectiveness.

### 2.4. Optimization of LCC and LCEI

In the context of building design optimization, MOO techniques have gained prominence as a response to the demand for sustainable and energy-efficient structures. Various MOO methods, such as evolutionary algorithms, Particle Swarm Optimization, and the WSA, have been applied in this area. The Non-Dominated Sorting Genetic Algorithm (NSGA) and other evolutionary algorithms in particular have become popular choices in the energy analysis of buildings.

Previous studies have focused on optimizing energy consumption and thermal comfort [32–36]. An effective strategy for optimizing green buildings involves addressing the high initial costs associated with Net Zeto Energy (NZE) buildings. Boermans et al. [32] and, subsequently, Xue et al. [33] examined LCC as it relates to energy consumption. The goal was to identify the cost-optimal solution. Wang et al. [34] employed genetic algorithms to find Pareto optimal solutions that meet both economic and environmental criteria. This approach offered a range of options, allowing for the selection of the most suitable solution based on specific preferences and priorities. Hong et al. [35] developed a complex mathematical model encompassing a vast number of building design scenarios. The Pareto optimal solutions were exceptionally well balanced; improving one objective would inevitably lead to the degradation of at least one other objective. Finally, Benincá et al. [36] compiled a comprehensive synthesis of 43 research articles employing MOO evolutionary algorithms. These articles are motivated by a multiplicity of goals: minimizing the effects of global warming, augmenting energy efficiency, enhancing user comfort, and analyzing the cost-effectiveness of optimized solutions across their entire life cycles.

However, the current research addresses a different set of objectives, necessitating a more efficient and streamlined approach to optimization. Given the precise objectives of this research, the WSA has been chosen for its advantages in situations where the number of iterations needs to be constrained for practical feasibility and efficiency. The WSA aggregates multiple objectives into a single objective function by assigning weights based on their relative importance, effectively guiding the optimization process toward favorable regions in the design space without requiring an exhaustive number of iterations. It allows for the consideration of trade-offs between various objectives without sacrificing the essential details of each criterion, facilitating stronger and more informed conclusions about sustainable building design.

This research drew inspiration from three previous studies: the first, by Islam et al. [37], focused on balancing LCC and LCEI in residential buildings in Australia. The study found the optimal envelope characteristics, reducing the environmental impacts without incurring additional costs. However, one limitation lies in its exclusion of occupant energy usage. Its

context differs from Ecuador, not only in terms of climate but also in material availability and construction practices, which vary between developed and developing countries. To enhance its research methodology, information from two more recent studies using the WSA was incorporated.

In a second study by Motuziené et al. [38], criteria weights were determined through a survey of experts, and the environmental impacts of three different types of building envelopes for a single-family house in Lithuania were assessed. The research employed LCA and LCCA to quantify the impacts associated with each envelope type and arrived at a decision using the WSA with the Analytic Hierarchy Process (AHP). The outcomes were sensitive to changes in criteria weights, highlighting the need for careful consideration of criteria prioritization.

The third study evaluated alternatives for a sustainable low-income housing design in Brazil [39], wherein Bianchi et al. encompassed a wide range of criteria, including costs, environmental impact, thermal comfort, construction duration, and cultural acceptance. Three different alternatives were effectively evaluated, assigning precise weights to each parameter based on questionnaires from both professionals and end-users. The methodology not only identified the optimal housing alternative but also emphasized specific phases and attributes that could benefit from refinement or integration.

## 3. Methodology

The primary object of this research methodology was to establish a clear, replicable, and pragmatic framework for future applications in the field of sustainable building design. The proposed methodology consisted of four stages:

1.    Selection of the single-family house and energy analysis;
2.    LCA with Tally;
3.    LCCA;
4.    MOO with the WSA.

This research followed a case study design, focusing on a specific single-family house, referred to as the base house (BH). The study's objective functions aimed to minimize both the LCEI and LCC of the house through a MOO approach. The BH was characterized by both constant and variable parameters, with the variables adjusted to create distinct alternative houses (AHs), each representing a unique iteration. The goal was to identify the most optimal solution among these alternatives. This research followed a quantitative approach, involving the collection and analysis of numerical data to establish relationships between the alternatives.

### 3.1. Stage 1: Selection of the Single-Family House and Energy Analysis

3.1.1. Base House

The BH model, initially created in Revit 2022, was used as the foundation for modeling the AHs in the same software. Constant parameters like location, orientation, geometry, and door/window placement remained fixed. Variable parameters were assigned for the envelope materials, led by various house iterations. The selection of the BH was based on the "Houses for Everyone" project, a national government initiative for social housing. Given the scale of this project, it was assumed that architectural and structural designs consider the availability of materials, construction methods, and regulatory requirements across the country, such as the chapters of the Ecuadorian Norm of Construction for seismic design [40]. The BH represented a conventional building, while the AHs simulated green building alternatives.

Revit was chosen for modeling given its comprehensive BIM capabilities, robust features for architectural design and documentation, and seamless integration with energy analysis and LCA tools like Autodesk Insight 2023 and Tally 2024. Data for modeling the BH were collected from official documents of the Ministry of Housing and Urban Development of Ecuador (MIDUVI). The BH architectural design caters to a typical Ecuadorian family, featuring three bedrooms, a living room, a kitchen, a bathroom, and a laundry area, with

a total surface of 55 square meters. The conventional construction method included a concrete foundation, steel profiles for the structure, and traditional masonry for the wall's envelope. For additional details, refer to Figure 1 and Table 1.

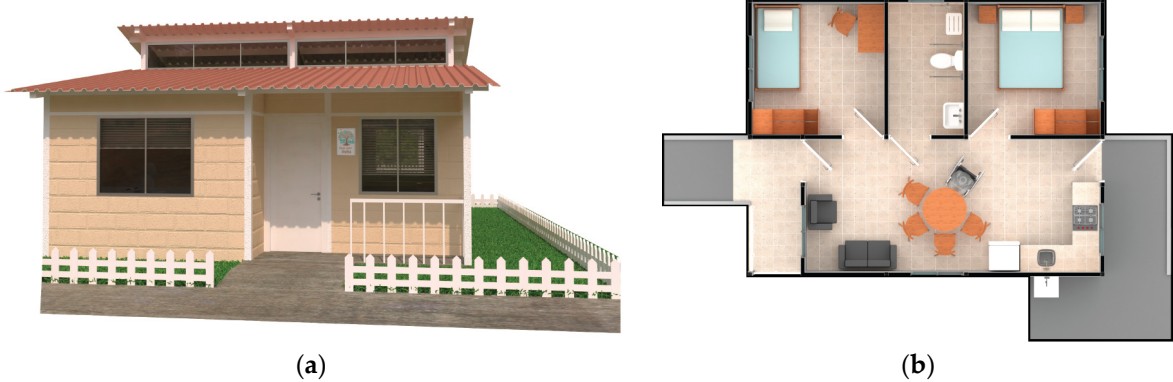

|       (a)       |        (b)        |

**Figure 1.** Base house: (**a**) 3D view; (**b**) plan view.

**Table 1.** Base house and alternative house materials. (Unit cost as of October 2023.)

| House | Specifications | | | |
|---|---|---|---|---|
| | **Material** | **Thickness** | **Life Duration** | **Unit Cost/m$^2$** |
| BH Wall | Concrete block and painting | 10 cm | 60 years | USD 20.04 |
| Wall 1 | Brick and sealant | 10 cm | 60 years | USD 19.23 |
| Wall 2 | Ecologic brick and sealant | 15 cm | 60 years | USD 26.81 |
| Wall 3 | Pine plank and varnish | 18 mm | 60 years | USD 56.00 |
| Wall 4 | Fir sandwich panel and varnish | 30 mm | 30 years | USD 56.34 |
| Wall 5 | Bamboo, borax, and linseed oil | 700 mm | 60 years | USD 23.54 |
| Wall 6 | Adobe and slaked lime | 20 cm | 60 years | USD 16.65 |
| BH Floor | Ceramic tiles and mortar | 11 mm | 60 years | USD 17.65 |
| Floor 1 | Epoxy Paint | 11 mm | 3 years | USD 12.00 |
| Floor 2 | Laminate flooring | 8 mm | 20 years | USD 21.32 |
| Floor 3 | Bamboo plank | 14 mm | 60 years | USD 38.39 |
| Floor 4 | Granite tiles and mortar | 15 mm | 60 years | USD 16.73 |
| Floor 5 | Stone tiles and mortar | 25 mm | 60 years | USD 45.23 |
| Floor 6 | Terracotta tiles | 17 mm | 60 years | USD 15.07 |
| Floor 7 | Polyethylene board | 5 mm | 20 years | USD 10.67 |
| Floor 8 | Polyaluminium board | 5 mm | 20 years | USD 11.35 |
| Floor 9 | Tetrapack board | 5 mm | 20 years | USD 10.00 |
| BH Roof | Galvalume | 12.4 mm | 40 years | USD 17.32 |
| Roof 1 | Wood teaks | 27 mm | 30 years | USD 54.70 |
| Roof 2 | Recycled cover | 6 mm | 20 years | USD 14.31 |
| Roof 3 | Clay tiles | 17.5 mm | 60 years | USD 43.69 |
| Roof 4 | Eternit cover | 5.5 mm | 30 years | USD 13.98 |

### 3.1.2. Alternative Houses

Selecting envelope materials as variable parameters was logical since these materials represent the second-largest volume in a house after structural materials. The structure remained a constant parameter given the need for precise structural design calculation, making alternative structural options less feasible. Variations in the AHs were achieved by treating the envelope assemblies—walls, floors, and roof—as variable parameters.

A list of AHs (Table 1) was established after considering material availability in the Ecuadorian market while bearing in mind the climate and cost restrictions. The decision was also informed by empirical knowledge and a review of the literature on green building practices. Some materials, like plastic recycled brick, were not included because of their

limited availability, despite being researched at some national universities [41,42]. To create the AHs, adjustments were made to the wall, floor, and roofing assemblies of the BH. Each iteration focused on altering one aspect of these assemblies at a time.

Detailed information on the selected materials was collected from commercial companies in Ecuador through direct communication between the authors and the providers. A Bill of Quantities (BOQ) was calculated for each of the materials, considering technical features like components and dimensions.

### 3.1.3. Energy Analysis

Autodesk Insight, a valuable building performance analysis program, enables the exploration of various metrics like energy use intensity, peak loads, and annual energy consumption. It enables the adjustment of parameters such as insulation, glazing, and PV system settings to compare simulation results and identify effective energy-saving strategies.

The BH was initially designed as an optimized passive house using Autodesk Insight, based on passive factors from a study by Sadeghifam et al. [43], which included considerations like energy efficiency, shading, night ventilation, lighting control, infiltration, and window size. These passive factors remained constant for all the alternative house designs, while the variable factors were the wall construction and roof construction. These changes were used to understand the impact of different envelope materials on energy consumption.

The results from Autodesk Insight provided energy consumption measurements in $kWh/m^2/yr$. It was observed that the wall material was the only parameter affecting energy consumption, producing different results for each alternative. In contrast, the roof and floor materials had a minimal impact on the energy consumption analysis.

For the BH and AHs, the energy systems relied on municipal electricity supply, with a specific electricity rate applicable in Ecuador as of October 2023 (0.092 Kw/h). Additionally, one of the AHs, called the PV House, was designed to incorporate photovoltaic panels. This design aimed to harness Ecuador's year-round solar availability, and in the case of the PV House, its energy consumption was assumed to be zero, reflecting an ideal scenario where the building generates as much energy as it consumes.

### 3.2. Stage 2: LCA with Tally

The estimation of the environmental impacts of the building elements was carried out using the Autodesk plug-in Tally 2024. Tally serves as a bridge between BIM elements in Revit and construction materials, drawing on its extensive database to establish connections and provide environmental impact assessments. Tally follows recognized environmental standards, like European Standard BS EN 159758 [44] and International Standard ISO 14040 [45], using environmental impact categories outlined in TRACI 2.1 [46]. LCA modeling is conducted using GaBi LCI (Life Cycle Inventory) databases, recognized worldwide in industrial and scientific applications [47].

In Tally, the assignment of materials from its database to each building element for analysis was a critical step, requiring careful consideration to ensure an accurate representation of the project's materials. Various parameters and settings within Tally were adjusted to match the specific context of the project, including factors such as geographic location, project type, construction phase, system boundaries, and functional units.

Tally also accounted for operational energy consumption, for which data from the Autodesk Insight analysis were manually input into the tool. Once all the necessary information was provided, Tally conducted the required calculations to perform the LCA and generated a comprehensive report that showcased environmental impact indicators for the analyzed building elements. This report was instrumental in assessing the environmental implications of the materials and designs used in the AHs.

### 3.3. Stage 3: LCCA

LCCA extends beyond the initial investment costs associated with construction, considering future expenses that arise during the operational and maintenance phases of a

building's life cycle. International Standard ISO 15686-5 [29] provides a structured framework for identifying and assessing the components, elements, and data required for a comprehensive LCCA [48].

In this case study, LCCA was a valuable tool for evaluating the economic performance of various housing options, including the BH and AHs. Equation (1) represents the mathematical expression used to compute the total life cycle cost, with an analysis period corresponding to the projected 60-year service life of the house. All monetary values were expressed in US dollars (USD).

$$LCC = I + M\&R + E \tag{1}$$

where

LCC—life cycle cost (USD);
I—investment costs (USD);
M&R—maintenance and repair costs (USD);
E—energy costs (electricity) (USD).

LCC calculations commonly employ the *NPV*, which converts the total costs associated with a project into a value equivalent to the present moment, considering the changing value of money over time [49,50]. *NPV* was calculated using Equation (2).

$$NPV = \sum_{i=1}^{n} \frac{CF}{(1+r)^t} \tag{2}$$

where

*NPV*—Net Present Value (USD);
*n*—number of years (year);
*CF*—cash flow (USD);
*r*—real discount rate (%);
*t*—time (year).

To calculate the *NPV*, the real discount rate was determined using the actual inflation rate and discount rate, as denoted in Equation (3). This discount rate accounts for the time value of money and allows for a fair comparison of costs incurred at different points in time.

$$r = \frac{(1+d)}{(1+i)} - 1 \tag{3}$$

where

*r*—real discount rate (%);
*d*—discount rate (%);
*i*—inflation rate (%).

In this context, inflation and deflation concepts were considered, and nominal costs reflected expenses estimated while considering the potential impact of price inflation or deflation over the project's life cycle. Nominal costs represent the actual amount of money that will need to be paid when the associated costs come due in the future [48]. It was assumed that all materials and electricity costs would experience the same rate of inflation.

The calculation of future inflation rates was based on historical data from periodic reports issued by the National Institute of Statistics and Census (INEC) [51] from 2004 to 2023. These reports provided insights into the consumer price index (CPI) for various categories of goods and services, allowing for an assessment of inflation trends. The average inflation rate was calculated for the last 20 years based on Ecuador's general inflation rate, and the result was 2.77%.

On the other hand, the discount rate converted future costs into their present values, considering the principle that money received or spent in the future is less valuable than money received or spent today. The discount rate, also known as the interest rate, is crucial

when comparing the LCC between different alternatives [48]. In this case study, the interest rate set by the Central Bank of Ecuador for social housing was 4.98% as of October 2023 [52].

By considering both inflation and discount rates, the LCCA provided a more accurate understanding of the financial implications of a project and facilitated the evaluation of its economic viability. These financial parameters are crucial when comparing the LCC of different housing alternatives.

### 3.4. Stage 4: Multi-Objective Optimization Method Weighted Sum Approach

The MOO approach used in this study employed the WSA to systematically evaluate a single-family house in Ecuador with the aim of minimizing both environmental impact and costs. The WSA assigned appropriate weights to different criteria or performance indicators, enabling the systematic prioritization and evaluation of alternatives [53]. The optimization process involved several steps, outlined in Figure 2.

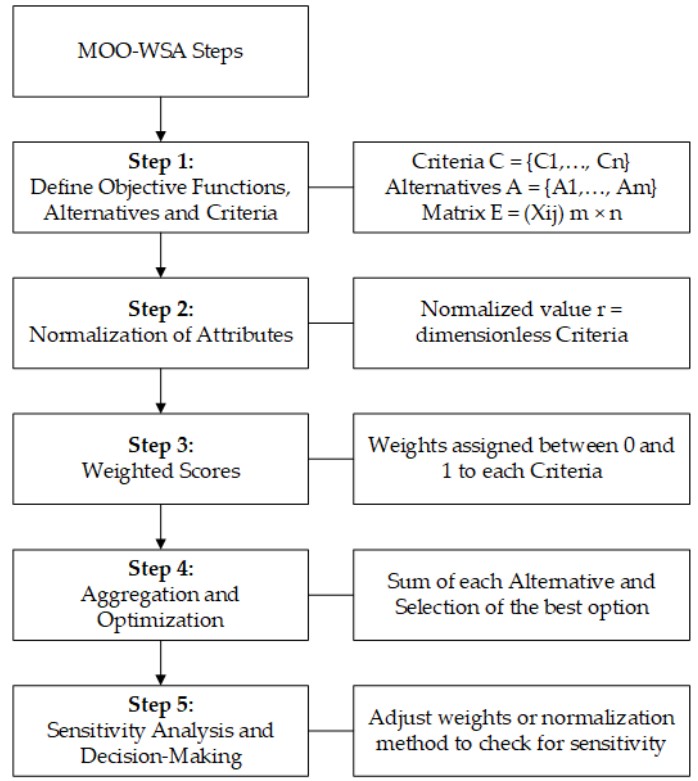

**Figure 2.** Steps in the multi-objective optimization with weighted sum approach.

The study focused on evaluating alternative envelope materials categorized into three groups: walls, floors, and roofs. A comprehensive analysis was conducted on a total of 20 alternatives, comprising 6 wall options, 9 floor choices, and 4 roofing materials, in addition to the BH. To simplify the process, the MOO-WSA was implemented three times, once for each category. The goal was to find the best result for each category separately in order to later combine these elements into a final optimized house (OH). Figure 3 describes the process for this optimization method, showing the solutions.

The top two alternatives from each category were obtained and combined to produce eight OHs, which are described in Table 2. Data for the LCA and LCCA were collected for the OHs in the same way as they were collected for the AHs.

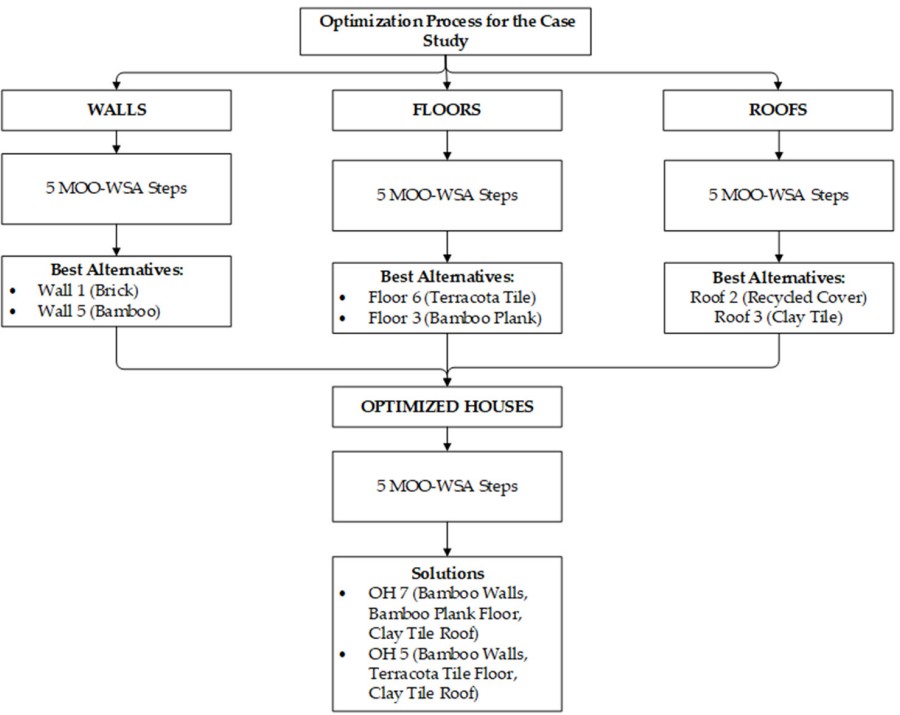

**Figure 3.** Optimization process for the case study.

**Table 2.** Optimized house materials.

| Optimized House Alternatives | | | |
|---|---|---|---|
| **House** | **Wall** | **Floor** | **Roof** |
| OH1 | Brick | Terracotta Tile | Recycled cover |
| OH2 | Brick | Terracotta Tile | Clay tile |
| OH3 | Brick | Bamboo Plank | Clay tile |
| OH4 | Brick | Bamboo Plank | Recycled cover |
| OH5 | Bamboo | Terracotta Tile | Clay tile |
| OH6 | Bamboo | Terracotta Tile | Recycled cover |
| OH7 | Bamboo | Bamboo Plank | Clay tile |
| OH8 | Bamboo | Bamboo Plank | Recycled cover |

### 3.4.1. Step 1: Define Objective Functions, Alternatives, and Criteria

In this initial step, the alternatives (walls, floors, roofs, and OHs) and criteria (LCC and LCEI) were defined. The criteria used for environmental impact included Primary Energy Demand (PED), Global Warming Potential (GWP), and Ozone Depletion Potential (ODP). These criteria were considered nonbeneficial, meaning that higher values indicated less preference.

### 3.4.2. Step 2: Normalization of Criteria

Normalization was applied to make the criteria comparable, transforming diverse measurement units into a common dimensionless scale ranging from 0 to 1, For nonbeneficial criteria, two different normalization methods were employed to allow for sensitivity analysis, as expressed in Equations (4) and (5).

Normalization method 1—linear scale transformation. This method transforms scores within a given criterion into proportions based on their relationship with the sum of all scores in that criterion.

$$r_{ij} = \frac{\frac{1}{x_{ij}}}{\sum \left( \frac{1}{x_{ij}} \right)} \tag{4}$$

where

$r_{ij}$—dimensionless criterion;
$x_{ij}$—evaluation of each alternative for each criterion (variable unit).

Normalization method 2—min–max. This method rescales values linearly within a specified range based on the minimum and maximum values of the range.

$$r_{ij} = \frac{max(x_{ij}) - x_{ij}}{max(x_{ij}) - min(x_{ij})} \tag{5}$$

### 3.4.3. Step 3: Weighted Scores

Assigning weights to the criteria was a critical aspect of the study. Previous studies have used various methods for weight assignment. Islam et al. [36] used a trial-and-error method, which, while straightforward, may lack a systematic approach. Motuziené et al. [37] and Bianchi et al. [38] used the Analytic Hierarchy Process (AHP), a structured method for determining the relative importance of criteria. The AHP involves a systematic process of pairwise comparisons where decision-makers compare criteria to each other and assign relative importance values.

The AHP starts by breaking down the problem into a hierarchal structure. Pairwise comparisons are made using the fundamental scale of Saaty, typically ranging from 1 to 9 [54], to express the relative importance or preference of one criterion over another. The AHP then synthesizes these comparisons to calculate the final weights for each criterion. It requires input from multiple stakeholders with relevant expertise.

For this study, the final weights were based on Motuziené et al. [37], where 30 experts with relevant backgrounds in energetics, environmental protection, and construction participated in a survey to determine the weights of PED, GWP, ODP, and LCC. Alternative weights from Bianchi et al. [38] were also considered, which included options like assigning equal weights to all criteria or giving the greatest weight to cost. The weight allocation is summarized in Table 3.

**Table 3.** Assigned weights for criteria.

| Criteria/Weight | Survey to Experts [37] | Equal Weights [38] | Greatest Weight Given to Cost (50%) [38] |
|---|---|---|---|
| Life Cycle Cost | 0.489 | 0.25 | 0.5 |
| Primary Energy Demand | 0.329 | 0.25 | 0.167 |
| Global Warming Potential | 0.119 | 0.25 | 0.167 |
| Ozone Layer Depletion | 0.064 | 0.25 | 0.167 |

### 3.4.4. Step 4: Aggregation and Optimization

The WSA combined the weighted scores for each objective by summing the products of normalized ratings and weights for each criterion. This process yielded the global value of each alternative $v(A_i)$. Alternatives were ranked in descending order based on their total values, with the alternative with the highest total value considered the most ideal choice and selected as the best option.

$$v(A_i) = \sum_{j=1}^{n} w_j r_{ij} \tag{6}$$

where

$v(A_i)$—global score of the alternative (dimensionless);
$w_j$—weighted score of the criterion (%);
$r_{ij}$—normalized value of each criterion in the alternative (dimensionless).

### 3.4.5. Step 5: Sensitivity Analysis and Decision Making

Sensitivity analysis was performed to ensure the robustness of the results and explore various trade-offs between environmental impact and cost. This involved adjusting the normalization methods and criteria weights to identify the most suitable design solution for balancing environmental impact and cost for the single-family house.

The research methodology was designed to offer a clear and replicable framework for future applications in this subject area, with a specific focus on addressing sustainability and cost considerations in single-family house design. Detailed information about the methods and software used, such as Autodesk Insight and Tally, are provided to allow others to replicate and build upon the results.

## 4. Results

The detailed results of the LCA and LCCA for each house alternative, along with the evaluation of relevant criteria, can be found in Table 4. Further analysis categorized the outcomes into walls, floors, roofs, and optimized houses. This separation allowed for easier comparisons of environmental impacts and life cycle costs. Additionally, the results of the optimization process are presented.

**Table 4.** LCA and LCC results for the houses.

| House | LCA | | | LCC (USD) |
|---|---|---|---|---|
| | Global Warming Potential (kgCO$_2$eq) | Ozone Layer Depletion (CFC-11eq) | Primary Energy Demand (MJ) | |
| Base House | 726,626.80 | $2.39 \times 10^{-4}$ | 13,671,900.55 | USD 25,288.07 |
| Wall 1 | 711,662.33 | $2.408 \times 10^{-4}$ | 13,381,486.29 | USD 24,991.38 |
| Wall 2 | 757,992.78 | $2.414 \times 10^{-4}$ | 13,817,078.94 | USD 25,946.46 |
| Wall 3 | 633,687.22 | $2.392 \times 10^{-4}$ | 11,959,942.11 | USD 58,895.47 |
| Wall 4 | 627,132.75 | $2.393 \times 10^{-4}$ | 11,877,658.98 | USD 58,938.65 |
| Wall 5 | 633,804.17 | $2.392 \times 10^{-4}$ | 11,949,294.42 | USD 31,334.52 |
| Wall 6 | 710,187.19 | $2.394 \times 10^{-4}$ | 13,163,741.67 | USD 30,932.16 |
| Floor 1 | 728,746.85 | $2.386 \times 10^{-4}$ | 13,782,530.95 | USD 30,301.28 |
| Floor 2 | 726,466.16 | $2.386 \times 10^{-4}$ | 13,682,797.25 | USD 26,382.93 |
| Floor 3 | 726,518.07 | $2.385 \times 10^{-4}$ | 13,674,099.29 | USD 26,138.41 |
| Floor 4 | 727,093.52 | $2.389 \times 10^{-4}$ | 13,678,945.58 | USD 25,250.35 |
| Floor 5 | 728,121.29 | $2.390 \times 10^{-4}$ | 13,694,952.06 | USD 26,418.85 |
| Floor 6 | 726,814.72 | $2.389 \times 10^{-4}$ | 13,675,013.07 | USD 25,182.29 |
| Floor 7 | 729,919.89 | $2.385 \times 10^{-4}$ | 13,743,894.34 | USD 25,474.53 |
| Floor 8 | 729,606.76 | $2.386 \times 10^{-4}$ | 13,735,692.06 | USD 25,532.53 |
| Floor 9 | 727,136.73 | $2.386 \times 10^{-4}$ | 13,685,322.79 | USD 25,417.38 |
| Roof 1 | 724,592.23 | $2.762 \times 10^{-6}$ | 13,693,989.09 | USD 54,437.35 |
| Roof 2 | 733,097.96 | $1.394 \times 10^{-6}$ | 13,812,307.78 | USD 25,707.66 |
| Roof 3 | 724,889.66 | $4.447 \times 10^{-6}$ | 13,664,714.21 | USD 27,828.76 |
| Roof 4 | 727,397.90 | $5.134 \times 10^{-6}$ | 13,672,984.84 | USD 25,009.95 |
| PV House | 5906.80 | $2.377 \times 10^{-4}$ | 89,100.55 | USD 28,900.81 |
| OH 1 | 18,721.69 | $2.148 \times 10^{-6}$ | 341,975.31 | USD 28,917.94 |

**Table 4.** *Cont.*

| House | LCA | | | LCC (USD) |
|---|---|---|---|---|
| | **Global Warming Potential (kgCO$_2$eq)** | **Ozone Layer Depletion (CFC-11eq)** | **Primary Energy Demand (MJ)** | |
| OH 2 | 10,156.10 | $5.114 \times 10^{-6}$ | 174,928.54 | USD 31,039.04 |
| OH 3 | 9888.64 | $4.752 \times 10^{-6}$ | 174,454.04 | USD 31,995.16 |
| OH 4 | 17,927.91 | $1.785 \times 10^{-6}$ | 329,837.37 | USD 29,874.06 |
| OH 5 | 1662.31 | $3.633 \times 10^{-6}$ | 49,546.17 | USD 37,382.18 |
| OH 6 | 9701.57 | $6.664 \times 10^{-7}$ | 204,929.50 | USD 35,261.08 |
| OH 7 | 1399.84 | $3.278 \times 10^{-6}$ | 49,080.52 | USD 38,338.30 |
| OH 8 | 9439.11 | $3.108 \times 10^{-7}$ | 204,463.85 | USD 36,217.20 |

### 4.1. LCA Results and Comparison by Categories

The environmental impacts encompassed three key criteria: Global Warming Potential (GWP) in kgCO$_2$eq, Ozone Layer Depletion (ODP) in CFC-11eq, and Primary Energy Demand (PED) in MJ. To facilitate comparisons, normalized values were used across different categories: walls, floors, roofs and OHs.

In the walls category, the results shown in Figure 4a unveil significant insights. Wall materials like pine, fir, and bamboo (Walls 3, 4, and 5) exhibited lower environmental impacts across all criteria, standing out for their eco-friendliness. In contrast, ecologic brick construction (Wall 2) displayed the highest environmental values. Adobe (Wall 6) mirrored the BH in having low ODP values but elevated GWP and PED values.

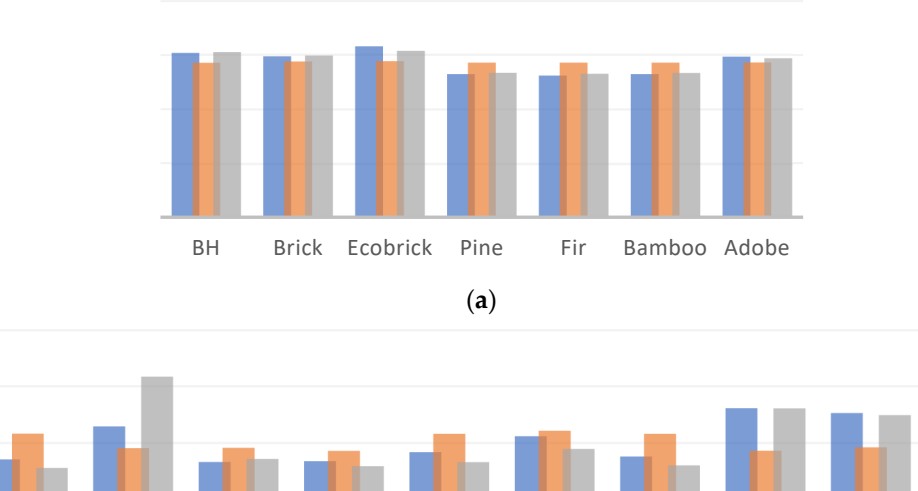

(**a**)

(**b**)

**Figure 4.** *Cont.*

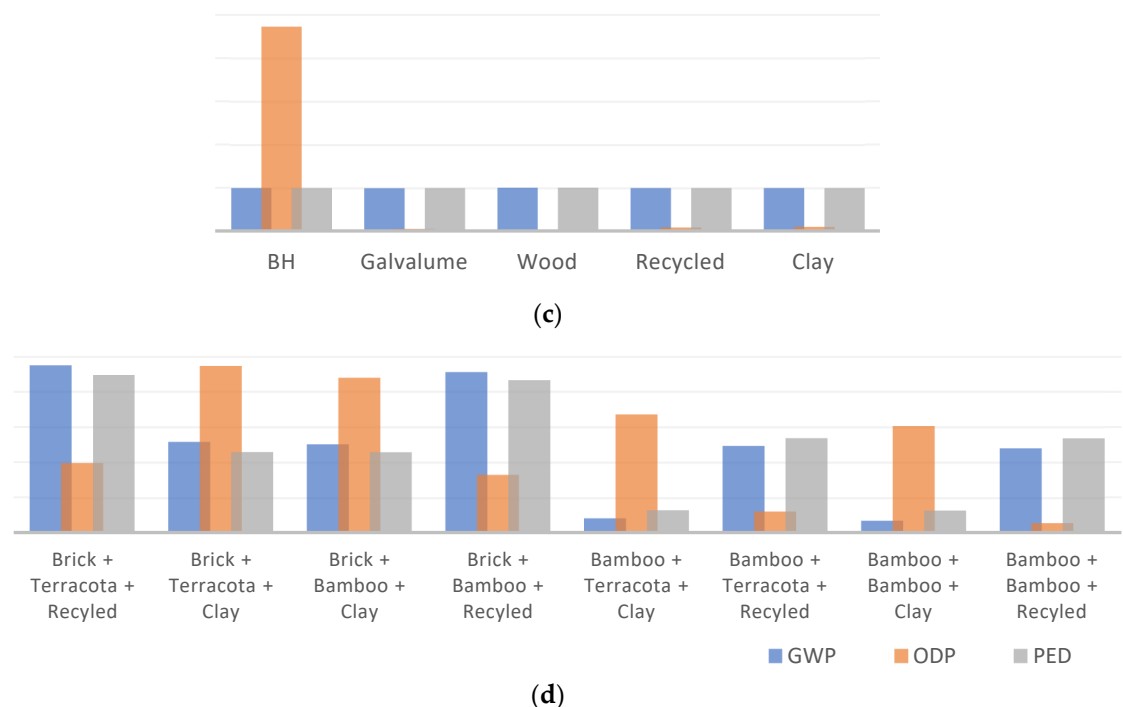

**Figure 4.** House LCA comparisons: (**a**) walls; (**b**) floors; (**c**) roofs; (**d**) optimized houses.

In the floors category, as shown in Figure 4b, the LCA result comparison revealed that the floors constructed with recycled materials, specifically, polyethylene and polyaluminium boards (Floors 7 and 8), had the highest GWP values. In contrast, wooden floors like laminate flooring and bamboo plank (Floors 2 and 3) demonstrated the lowest GWP values.

Regarding ODP, the BH shared high values with stone, granite, and terracotta tile floors (Floors 4, 5, and 6), while wooden floors and recycled boards showed lower ODP values. When considering PED, epoxy paint (Floor 1) incurred the highest impact, with Floors 7 and 8 following closely. Ceramic tiles, bamboo planks, and terracotta tiles (BH, Floors 3 and 6) had the lowest PED impact.

In the roofs category, shown in Figure 4c, the BH exhibited notably high ODP compared with the AHs, with Roof 2 (composed of recycled polyaluminium) displaying the lowest ODP value. For the other criteria, values remained relatively consistent among the alternatives. Notably, Roof 1 (featuring wood teak) had the lowest GWP among all the roofs, emphasizing its eco-friendly nature. Roof 3 (constructed with clay tiles) boasted the lowest PED, signifying its energy-efficient attributes.

Lastly, the analysis of the OHs revealed noteworthy findings, as shown in Figure 4d. Houses with bamboo walls and clay tile roofing—differentiated by their flooring material (OH5 and OH7), one with terracotta tiles and the other with bamboo plank—exhibited the lowest impacts in terms of GWP and PED. Conversely, OH1 and OH4, both featuring brick walls and recycled cover roofing, demonstrated the highest impacts in terms of these criteria. For ODP, the houses with the lowest impact were OH6 and OH8, with bamboo walls and recycled cover, differing in the floor type. OH2 and OH4, featuring brick walls and clay tile roofing, exhibited the highest ODP impact.

These insights provide a comprehensive understanding of how various construction materials impact the environment and pave the way for selecting the best alternatives in the optimization process.

### 4.2. LCCA Results

In the data collection phase of the optimization process, the LCC was calculated for the BH and AHs. Figure 5a provides an overview of their results. Most designs exhibited

similar costs with the BH, falling within a range of USD 25,000 to USD 30,000. However, three alternatives stood out with significantly higher costs. Walls 3 and 4, both constructed with wooden materials (pine and fir), had costs of nearly USD 60,000. Roof 1, which also featured teak wood, was the third most expensive design. Among the floor alternatives, Floor 1, composed of epoxy paint, was the most expensive.

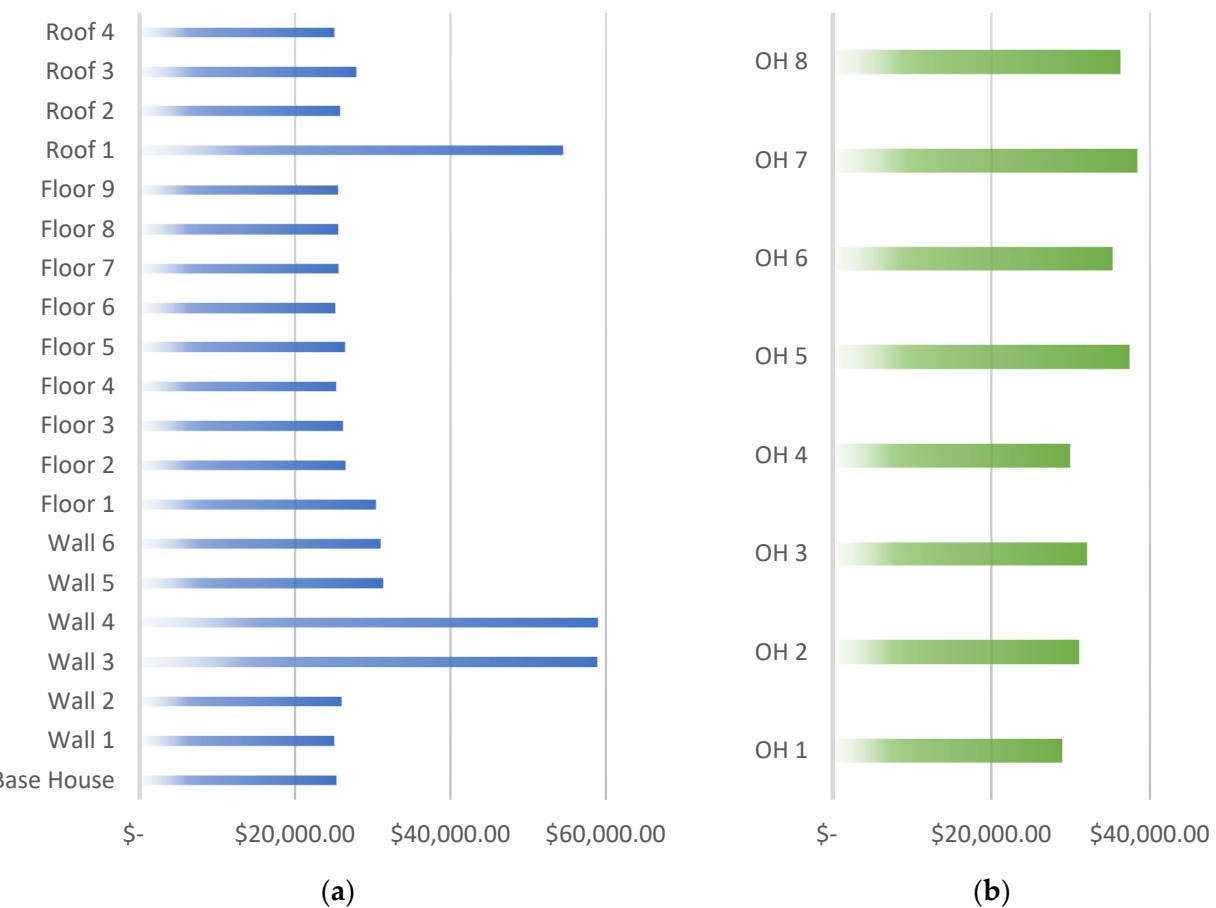

**Figure 5.** House LCC: (**a**) alternative houses; (**b**) optimized houses.

Subsequently, the optimization process analyzed the LCC of the OHs depicted in Figure 5b. Their costs ranged from approximately USD 30,000 to USD 40,000. An important consideration for the OHs was the optimization of their energy systems. This entailed calculating the cost of energy consumption during their lifespans using PV systems rather than grid-supplied electricity, which was slightly more expensive. The OHs' LCC exhibited a distinct pattern, with OH1–OH4 (comprising brick walls) having an approximate cost of USD 30,000. OH5–OH8 (constructed with bamboo walls) had costs ranging from USD 30,000 to USD 38,000. OH1, composed of brick walls, terracotta tile flooring, and recycled cover roofing, emerged as the most cost-effective alternative.

### 4.3. MOO-WSA Results

Following the optimization process outlined in Figures 2 and 3, the study arrived at the best alternatives for each house design category. Figure 6 presents the global scores for each alternative, $v(A_i)$. During the sensitivity analysis, it was discovered that the results differed when the criteria were normalized using either method. As a result, the study identified the two best alternatives for each category, leading to two optimal solutions at the conclusion of the optimization process.

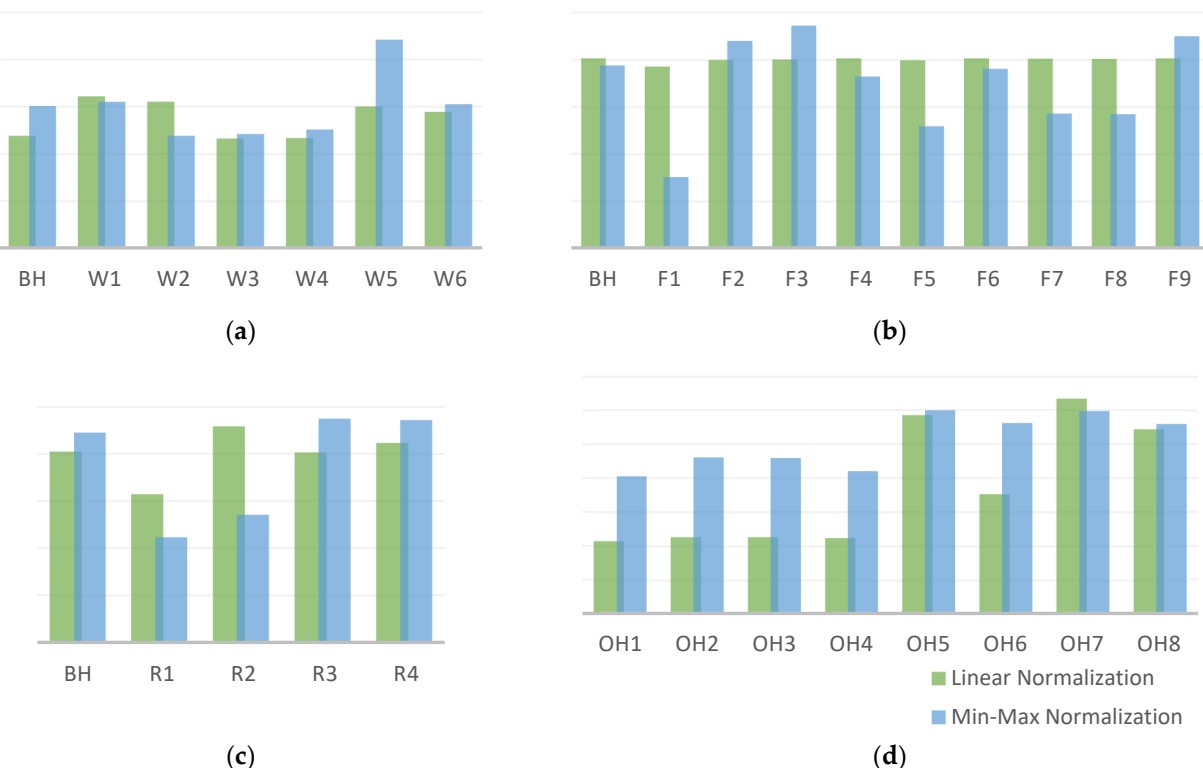

**Figure 6.** House global scores: (**a**) walls; (**b**) floors; (**c**) roofs; (**d**) optimized houses.

In the first category, walls, the best alternative found using linear-scale normalization was Wall 1, composed of bricks. The best alternative found using min-max normalization was Wall 5, made of bamboo. For the floors category, linear-scale normalization indicated that Floor 6, simulated with terracotta tiles, was the best alternative, while min-max normalization showed that Floor 3, composed of bamboo, was the superior choice. In the roofs category, under linear-scale normalization, Roof 2, consisting of recycled cover materials, emerged as the best alternative, and under min-max normalization, Roof 3, composed of clay tiles, was deemed the optimal choice. Lastly, in the OH category, the best final solutions were determined. Using linear-scale normalization, OH7, combining bamboo walls and floors and clay tile roofing, was identified as the best solution. On the other hand, using min-max normalization, OH5, which closely resembled OH7 but featured terracotta tiles flooring, was designated as the superior choice. We called the last solutions sustainable houses. Figure 7 represents the Revit model for the OH featuring bamboo walls and clay tiles roofing.

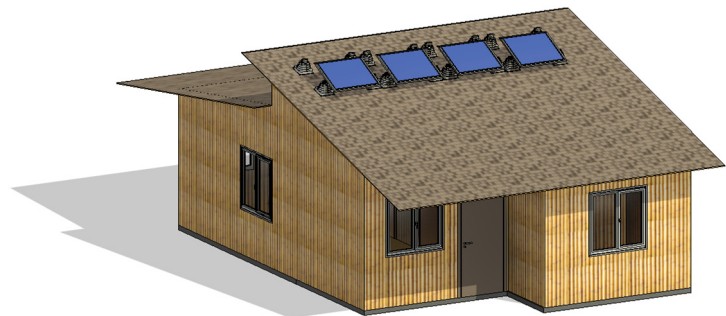

**Figure 7.** Sustainable house model in Revit.

*4.4. Sustainable House Solutions*

The primary objective of this research was to identify a superior alternative for the BH that would replace its envelope materials, resulting in a sustainable yet cost-effective

design. OHs OH7 and OH5 successfully achieved this goal, showcasing the effectiveness of the proposed optimization process. Table 5 summarizes the rate of change between the BH and the optimized solutions.

**Table 5.** Rate of change between the sustainable houses and the base house.

| House | Global Warming Potential | Ozone Layer Depletion | Primary Energy Demand | LCC |
|---|---|---|---|---|
| OH5 | −72% | −98% | −44% | +29% |
| OH7 | −76% | −99% | −45% | +33% |

The OHs demonstrated exceptional performance in the LCA criteria. Specifically, they achieved a remarkable 75% reduction in GWP, an astounding 99% decrease in ODP, and a significant 45% reduction in PED. Nonetheless, it is worth noting that the sustainable house solutions still incurred a LCC approximately 30% higher than the BH.

Finally, the research findings suggest that it is possible to design single-family houses with significantly reduced environmental impacts through the optimization of envelope materials. The OH7 and OH5 designs exemplify the effectiveness of this optimization process. However, it is crucial to strike a balance between these environmental benefits and the increased cost associated with sustainable construction materials, as this can have financial implications for housing projects. These results provide valuable insights for architects, engineers, and decision-makers seeking to find the right equilibrium between sustainability and cost-effectiveness in residential construction.

## 5. Discussion

This research tackles the challenges of sustainable construction, with the aim of transforming an affordable single-family house into an economically viable and sustainable dwelling through a practical optimization process applicable to real-world projects. Promoting green housing alternatives in Ecuador contributes to improved living conditions and the preservation of the country's remarkable biodiversity.

The successful optimization used MOO and the WSA. The focus was primarily on envelope materials including walls, floors, and roofs. The BH featured block walls, ceramic tile flooring, and galvalume roofing, whereas, after the optimization, the sustainable houses featured bamboo walls, terracotta and bamboo plank flooring (two options), and clay tile roofing.

Although bamboo material may seem innovative and potentially risky, it was successfully used in the construction of 108 bamboo houses in Ecuador in 2020 as part of the "Houses for Everyone" project [55] (Figure 8). This demonstrates the practicality and acceptance of the proposed solution among local builders, reinforcing the feasibility of green housing alternatives.

Consistent with a previous study by Soust-Verdaguer et al. [56], this study showed that the most substantial environmental impacts during a construction's life cycle occur across the operational phase because of energy consumption, as well as during maintenance and replacement activities. These findings are instrumental in the sustainability approach. To address operational energy consumption, the strategic choice of a PV system was implemented. This decision reduced the carbon footprint and mitigated environmental harm.

The LCA of the OHs shed light on the behavior of floor and roof materials. It became evident that floors had minimal impact on the LCA because of their relatively small volume in comparison with the other two assemblies. Conversely, roofing materials showed an interesting contrast in their effect on GWP and PED compared with ODP. Clay tile roofing resulted in high values in the ODP criterion but low values in GWP and PED, while the recycled cover displayed the opposite performance. This divergence could be attributed to the fact that clay tiles are made from natural clay, making them friendly in terms of greenhouse gases (GHGs) and embodied energy, but their production includes minerals and oxides

that affect the ozone. In contrast, recycled materials made of aluminum typically involve $CO_2$ emissions but do not contain substances with ODP, like chlorofluorocarbons (CFCs).

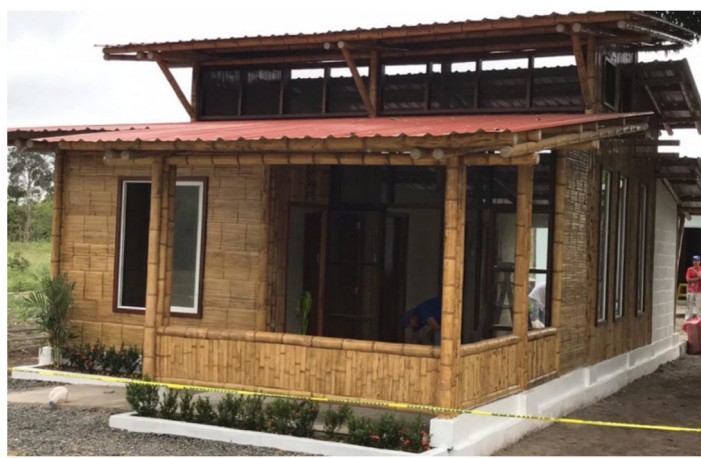

**Figure 8.** "Houses for Everyone" project: bamboo house.

The LCCA also yielded valuable insights. It was found that the most expensive materials were those composed of wood, except for the floor. Wood-based materials for walls and roofs require constant maintenance to ensure their durability and good condition, as wood can be very sensitive to deterioration. Despite the higher cost, the analysis revealed that wood-based components displayed the lowest values across the spectrum of environmental impacts, which aligns with the findings of a previous study by Soust-Verdaguer et al. [56] regarding the environmental advantages of utilizing wood.

The optimization process relied on two different normalization methods: the linear-scale method and the min-max method. The linear-scale method produced a dataset with fewer variations and more average values, while the min-max method highlighted differences in the data by emphasizing the highest and lowest values. Determining the most accurate or appropriate method proved challenging, resulting in two solutions at each step and two final solutions for sustainable house designs.

It is worth noting certain limitations of this research. First, there was a limitation in modeling ecologic bricks and adobe walls using the Tally data inventory. To address this issue, an approximation was made by modeling them as concrete mixtures replacing cement with lime. While this was a practical solution, it did not fully capture the unique characteristics of those materials, reflected in the high values for their LCA criteria. This limitation prompts the need for a broader and more inclusive approach to the data inventories of Tally 2024 LCA software. Different kinds of materials should be included, encompassing those rooted in ancient building techniques like adobe and bahareque, as well as modern materials like recycled bricks from PET bottles. While these were not part of this study, their inclusion is essential to gaining popularity within the construction industry and architectural design tools.

Another limitation was the inherent uncertainties in the LCCA process. LCCA hinges on various assumptions and data inputs, some of which may introduce elements of unpredictability into the analysis. For instance, the reliance on lifespan data provided by commercial companies and the assumption of uniform inflation rates for all materials and energy usage can introduce variability into the analysis.

These limitations suggest that further sensitivity analysis could be implemented in earlier stages of optimization rather than only in the normalization or weighting steps. In the case of the Tally data inventory, we recommend conducting an LCA using other programs with different inventories from GaBi. Additionally, making a deeper investigation into material providers would be beneficial, allowing the designer to compare prices and technical details such as durability and dimensions.

Finally, the sensitivity analysis may also vary the material selection in the first stage when considering climate change or earthquake exposure. Climate change in Ecuador can introduce extreme weather events like droughts and heavy rainfalls, which may necessitate the use of durable and weather-resistant materials. The maintenance and replacement requirements should be adjusted accordingly. For earthquake exposure, materials should be designed following the structural requirements for earthquake resistance [40] or the specific norm for bamboo construction in Ecuador [57].

## 6. Conclusions

In the quest to transform the BH into sustainable houses, several alternative house designs were explored, and several conclusions can be drawn from the LCA, LCCA, and optimization stages.

In commitment to sustainability, the durability of materials is of paramount importance. Materials with longer lifespan durations eliminate the need for frequent replacements, thereby reducing overall environmental impacts and replacement costs. This is exemplified by the case of recycled floors. Despite being made from salvaged materials, they showed significant results for LCA criteria. This could be attributed to their relatively short lifespan of 20 years, necessitating two renewals during the building's life cycle.

The eco-friendly nature of wood makes it an attractive material for sustainable house design. Bamboo emerged as the best solution given its cost-effectiveness. Bamboo requires different maintenance products compared with other wooden materials analyzed in this study (fir and pine), which are less expensive but environmentally harmful. However, it still accounts for intensive maintenance costs. Therefore, it is essential to note that the sustainable houses increased the LCC of the BH by 30%.

Despite the cost increase, the sustainable houses demonstrated a tremendous positive impact on environmental conservation. By embracing eco-friendly materials and a sustainable energy system, the environmental impacts were significantly reduced. They exhibited a stunning 98% decrease in ODP; a 75% reduction in GWP; and a substantial 45% drop in PED.

A promising direction for future research would be to find a balance between green and conventional materials so that the house does not experience a 30% increase in costs. In other words, it means calculating the percentage of the house materials that should become green (bamboo, clay) and the percentage that should remain conventional (concrete blocks, metal). This approach may not result in the same level of decrease in environmental impacts, but it would be more attractive for constructors and owners to maintain their budgets while still reducing their carbon footprints.

Moreover, it would be interesting to include structural materials in the analysis, recognizing the critical role of the structure in shaping a building's sustainability, especially as it often constitutes the largest volume of materials used in construction. For instance, when using bamboo in walls, there might be an opportunity to explore bamboo as a structural material as well. This holistic approach not only aligns with sustainability goals but also ensures the integration of all building elements.

An intriguing aspect of the research journey was the revelation of the undeveloped state of the green construction industry in Ecuador. In contrast to other countries in the region, like Brazil, Colombia, and Mexico, where a broader spectrum of ecological materials is readily available, Ecuador appears to have limited options for recycled materials, with only one company found to be offering products like recycled cover roofing and recycled board flooring. This lack of diversity and competition in the market underscores the need for concerted efforts to stimulate the growth of the green construction sector in Ecuador.

We expect the present study to inspire engineers, architects, and designers to embrace optimization tools and contribute to transforming the construction industry toward more environmentally responsible practices. The research inputs encourage designers and end-users to opt for sustainable housing by using a refined and quantitatively validated model. Furthermore, it bridges academia and the construction industry by applying tools like BIM

and LCA, incorporating green building material concepts, and demonstrating the economic and environmental benefits of sustainable construction. However, driving widespread change in the AEC industry necessitates a collaborative endeavor involving policymakers and regulators and shared responsibility from all stakeholders.

The journey toward sustainability in construction is a collective effort, demanding both academic investigations and industrial production. By addressing these challenges and embracing opportunities for improvement, the study charts a path toward a greener, more sustainable future in construction. It reflects a commitment to environmental conservation and responsible construction practices.

**Author Contributions:** Conceptualization, Y.C. and S.G.; methodology, S.G.; software, S.G.; validation, Y.C. and S.G.; formal analysis, S.G.; investigation, S.G.; resources, S.G.; data curation, S.G.; writing—original draft preparation, S.G.; writing—review and editing, Y.C.; visualization, S.G.; supervision, Y.C.; project administration, Y.C.; funding acquisition, Y.C. All authors have read and agreed to the published version of the manuscript.

**Funding:** This research received no external funding.

**Institutional Review Board Statement:** Not applicable.

**Informed Consent Statement:** Not applicable.

**Data Availability Statement:** The data presented in this study are available on request from the corresponding author.

**Conflicts of Interest:** The authors declare no conflict of interest. The funders had no role in the design of the study; in the collection, analyses, or interpretation of the data; in the writing of the manuscript; or in the decision to publish the results.

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
