# Peer review of "A Multi-Objective Optimization Method for the Design of a Sustainable House in Ecuador by Assessing LCC and LCEI"

_sustainability, doi:10.3390/su16010168_

Round 1
Reviewer 1 Report
Comments and Suggestions for Authors
General Comments
This paper presents an accurate and comprehensive sustainability analysis of low income housing in Ecuador, with interesting results and increased insight into optimized materials and configurations with respect to energy environment and costs on the life span.
Nevertheless, authors could add few comments on few factors that can influence the sensitivity analysis, such as 1) climate change induced increased frequency of extreme events; and 2) earthquake exposure.
Since the BH and other configurations are built based on a national program, and Ecuador is a highly seismic country, I am sure earthquake design has been accounted for at the collapse prevention or life safety level, for these simple structural types (steel frames, as the authors mentioned). However since results point out that sustainable material choice may impact the cost analysis, some comments should be added on the likelihood that frequent low-to-moderate seismic event affect serviceability of components, thus replacement and repair costs. The literature has now a number of recent joint approaches, accounting for both environmental performance and hazard (earthquake and others) performance.
Other Comments
GHG ? not defined
Line 652 revise sentence
Fig 4 a and b there is no metric scale, revise
Fig 4 add names of configuration in the scales (to see what each configuration features), instead of abbreviations, for ease of reading results
Fig 5 add metric
Author Response
Dear Reviewer,
Please see the attatchment where all the comments are solved as follows.
Line 161: background on climate change
Line 301: Background on seismic design
Paragraph starting in line 724: comments on sensitivity analysis discussing climate change and earthquake exposure
Line 684: GHG detailed
Line 652 changed to 744: changed
Fig 4 and 5: changed
I look forward to receiving additional recommendations and addressing any further concerns you may have.

Reviewer 2 Report
Comments and Suggestions for Authors
The manuscript sustainability-2736922, titled “A Multi-Objective Optimization method for the design of a sustainable house in Ecuador by assessing LCC and LCEI”, assessed the cost-effectiveness of sustainable materials, focusing on envelope materials in Ecuador. The complicated relationship between the interacted components is explained in detail step by step. The research is interesting, innovative and powerful, supplying inspirational insights into the solutions to tackle similar issues. This manuscript satisfies the research scope of Sustainability, but major revisions are required to improve the manuscript’s quality.
1. Abstract:
1) The authors are from China,why they are focusing on envelope materials in Ecuador?
2) Quantitative results do not effectively express results from the study, so the most important results obtained must be numerically mentioned.
2. Introduction:
3) There are many sentences in the text that have errors in grammar and should be corrected. The authors suggest doing a proof of English reading and editing a manuscript to correct all grammar errors.
3. Ecuadorian Context
4) I don't think a temperature of 23 to 26 degrees causes a problem for the surrounding environment, construction material and design considerations. The authors should comment in the text.
4. Optimization of the LCC and LCEI
5) Findings of references from 28-35 should be discussed in the text.
5. Methodology
6) On what basis do the authors consider the range or level of each of the studied cast studies. It must be discussed in the manuscript.
7) In Table 1, the unit cost is variable with the time. The authors should specify the month and the year when this cost was calculated.
8) How can the results of this paper help to improve the construction industry?
9) The findings of this research should be compared with previous results.
10) Units of Equations should be added.
6. Conclusion:
11) Discussions and conclusions should be separated.
12) The conclusion section should be re-written to include specific general points. The conclusion in its current form is considers as a discussion.
Comments on the Quality of English Language
Minor editing of English language required
Author Response
Dear Reviewer,
Thank you very much for your comments. Please see the attachment with all the responses as follows.
1) The authors are from China, why they are focusing on envelope materials in Ecuador?
The second author is from Ecuador, with professional experience in the construction of the houses from the case study.
2) Quantitative results do not effectively express results from the study, so the most important results obtained must be numerically mentioned.
Lines 16-18
3) There are many sentences in the text that have errors in grammar and should be corrected. The authors suggest doing a proof of English reading and editing a manuscript to correct all grammar errors.
Many mistakes were corrected, please let us know if there are still issues regarding this matter.
4) I don't think a temperature of 23 to 26 degrees causes a problem for the surrounding environment, construction material and design considerations. The authors should comment in the text.
The temperature does not "influence" the construction materials as is was described before, but it defines the range of options. Line 159.
5) Findings of references from 28-35 should be discussed in the text.
Paragraph starting in line 229
Lines 255-258
Lines 264-268
Lines 272-275
6) On what basis do the authors consider the range or level of each of the studied cast studies. It must be discussed in the manuscript.
Lines 323-324
7) In Table 1, the unit cost is variable with the time. The authors should specify the month and the year when this cost was calculated.
Line 315
8) How can the results of this paper help to improve the construction industry?
Paragraph starting in line 772.
9) The findings of this research should be compared with previous results.
Paragraph starting in line 669.
10) Units of Equations should be added.
Added accordingly to each equation.
11) Discussions and conclusions should be separated.
Done
12) The conclusion section should be re-written to include specific general points. The conclusion in its current form is considers as a discussion.
Done
I look forward to receiving additional recommendations and addressing any further concerns you may have.

Reviewer 3 Report
Comments and Suggestions for Authors
Dear Authors,
I am writing to provide feedback on your manuscript. Your research, employing a multi-objective optimization approach and weighted sum approach, is an essential contribution to sustainable construction practices. However, there are some areas where further development could enhance the overall impact of your study.
1. How was the accuracy and representativeness of the data used in LCA and LCCA ensured? Could further sensitivity analysis be added to assess the robustness of the findings?
2. It would be beneficial to include a background paragraph on different methodologies related to construction process improvement in similar contexts. Mentioning methods beyond LCA, such as Material Flow Analysis (MFA), Lean Construction, and Building Information Modeling (BIM), might provide new insights. In this sense, to enrich your background and the benefits of your research, please include the following studies from the Andean region and highlight the implications of your study in the current context:
BIM Adoption among Contractors: A Longitudinal Study in Peru. https://doi.org/10.1061/(ASCE)CO.1943-7862.0002424
Qualitative Analysis of Lean Tools in the Construction Sector in Colombia. https://doi.org/10.24928/2019/0185
3. Have more comprehensive environmental impact categories, such as water usage and waste generation, been considered?
4. Address the potential scalability challenges and opportunities of your study. What are the practical applications of your study in current and future housing projects in Ecuador and similar countries?
5. Include the main limitations of your current study and propose future research directions, perhaps involving the use of renewable energy sources or the integration of complementary methodologies, would provide a more wide-ranging scope.
Your manuscript provides significant contributions to the field of sustainable housing design, particularly in Ecuador. Incorporating the above suggestions will enhance the practical applicability of your work.
Author Response
Dear Reviewer,
Thank you very much for your valuable comments. Please see the attachment to find all the responses as follows.
1. How was the accuracy and representativeness of the data used in LCA and LCCA ensured? Could further sensitivity analysis be added to assess the robustness of the findings?
Paragraph starting in line 330.
Lines 364-365.
Paragraph starting in line 718.
2. It would be beneficial to include a background paragraph on different methodologies related to construction process improvement in similar contexts. Mentioning methods beyond LCA, such as Material Flow Analysis (MFA), Lean Construction, and Building Information Modeling (BIM), might provide new insights.
The articles were carefully analyzed, and the recommended paragraphs were introduced in the Background section with the title "Environmental Analysis" from line 81.
3. Have more comprehensive environmental impact categories, such as water usage and waste generation, been considered?
These parameters were not considered in the environmental analysis since the used tool (LCA with Tally) is not able to calculate these categories. However, it could be interesting for a future research direction.
4. Address the potential scalability challenges and opportunities of your study. What are the practical applications of your study in current and future housing projects in Ecuador and similar countries?
Paragraph starting in line 772.
The methodology can be replicated in any kind of project with similar weather conditions.
5. Include the main limitations of your current study and propose future research directions, perhaps involving the use of renewable energy sources or the integration of complementary methodologies, would provide a more wide-ranging scope.
Paragraph starting in line 702, 713 and 718.
Paragraphs starting in line 752 and 759.
It is worth mentioning that the study makes an integration of four methodologies: BIM, LCA, LCCA and WSA. What is more, it also includes PV (photovoltaic) panels for the energy system, as a renewable energy source.
I look forward to receiving additional recommendations and addressing any further concerns you may have.

Round 2
Reviewer 2 Report
Comments and Suggestions for Authors
Now the paper can be accepted
Author Response
Thank you very much
Reviewer 3 Report
Comments and Suggestions for Authors
Thank you for your answers. I believe that the article is much better articulated now. Still, please check reference 12. It should be:
Castiblanco, F.M., Castiblanco, I.A., Cruz, J.P. Qualitative analysis of lean tools in the construction sector in Colombia (2019). 27th Annual Conference of the International Group for Lean Construction, IGLC 2019, pp. 1023-1036. doi: 10.24928/2019/0185
Author Response
Dear Reviewer,
Reference 12 has been changed accordingly.
Thank you for your suggestions.
